# Energy evolution mechanism of structural surfaces in sandstones with different dips based on the energy principle

**Yongjiang Yu, Zhiyuan Song** [ID]*, **Jiaming Liu, Yuntao Yang, Xu Dong**

College of Mining, Liaoning Technical University, Fuxin, China

* 1712828821@qq.com

**Data Availability Statement:** All relevant data is contained in the manuscript and its Supporting information file.

**Funding:** The author(s) received no specific funding for this work.

## Abstract

A uniaxial compression test was conducted on sandstone specimens at various inclination angles to determine the energy evolution characteristics during deformation and damage. Based on the principle of minimum energy dissipation, an intrinsic model incorporating the damage threshold was developed to investigate the mechanical properties of sandstone at different inclination angles, and the energy damage evolution during deformation and damage. This study indicated that when the inclination angle of the structural surface remained below 40˚, sandstone exhibited varying mechanical properties based on different inclination angles. The peak strain was positively correlated with the inclination angle, whereas the compressive strength and modulus of elasticity showed negative correlations. From an energy perspective, the deformation and damage of sandstone under external loading entail processes of energy input, accumulation, and dissipation. Moreover, higher inclination angles of the structural surface resulted in a smaller absorbed peak strain and a reduced proportion of dissipated energy relative to the energy input, thereby affecting the evolution of energy damage throughout the process. As the inclination angle of the structural surface increased, the absorbed total strain at the peak value decreased, whereas the proportion of the dissipated energy increased. Additionally, the damage threshold and critical value of the rock specimens increased with the inclination angle. The critical value, a composite index comprising the peak strain, compressive strength, and elastic modulus, also increased accordingly. These findings can offer a novel perspective for analyzing geological disasters triggered by fissure zones within underground rock formations.

## 1 Introduction

As mining technology advances, underground excavation extends deeper, with the emergence of various large-scale joints and fissure zones within rock body [1]. The rock mass stability can be affected differently by the inclination angles of the structural surfaces under various construction conditions. When subjected to external loads, a rock mass containing structural surfaces may experience collapse, sheet fractures, and other geological disasters [2–4]. Therefore, investigating the deformation, damage characteristics, and energy evolution mechanism of

**Competing interests:** The authors declare that they have no conflicts of interest.

rock masses with diverse inclination angles holds significant theoretical importance for exploring disaster prevention and control measures in underground settings [5].

Globally, scholars have conducted numerous studies employing uniaxial and triaxial compression tests. Zhang Xiaowu [6] analyzed the damage characteristics and energy evolution laws in six types of rocks with varying inclination angles using uniaxial compression tests. Qi Xiaohan [7] examined energy evolution in coal samples before and after peak stress via multiple compression tests. Xie Heping [8, 9] elucidated the energy damage evolution mechanism in rocks with different strengths. Xu Wentao [10] conducted triaxial compression tests on easily ejected coal bodies, analyzing the transformation of elastic potential energy and dissipation energy during destruction. Zhang Wei [11] performed uniaxial compression experiments on prefabricated cement specimens with different inclination angles, observing the smallest compressive strength occurred at a 45˚ inclination angle under static loading conditions. Li Shuchen et al. [12] systematically summarized the impacts of fissure inclination angles on key physical parameters (post-peak stress-strain curves, residual strength, and Poisson's ratio) through conducting uniaxial compression tests on prefabricated jointed rock specimens. Yang et al. [13] constructed a mechanical model of jointed rock fragmentation and studied the impact of the joint inclination angle size on the degree of rock fragmentation. Sun Mengcheng [14] developed a new constitutive model for rock damage through applying the principle of minimum energy consumption. Yang Dayong [15] proposed a novel concept of rock damage evolution using the theory of minimum energy consumption, deriving the damage evolution equation and damage critical value of rock blocks via uniaxial compression experiments. Although these experts have extensively explored the damage energy characteristics of rocks with various inclination angles, the impact of the damage critical value has received limited attention. Hence, this study analyzed the energy evolution mechanism of different structural surface inclination angles adopting the theory of minimum energy dissipation, established an intrinsic damage model for rock bodies under different structural surface inclinations, and examined the structure of the rock body and the evolution law of strain energy release rates of damage based on this foundation.

## 2 Mechanical properties of rock body with different inclination structural surfaces

### 2.1 Stress-strain curves of rock body with different inclined structural surfaces

To investigate the energy change patterns during the damage process of structural surfaces with varying inclinations, uniaxial compression tests were performed on rock specimens at the Mechanics Laboratory of the Liaoning University of Engineering and Technology. The tests utilized an MTS815 all-digital hydraulic servo test system (Fig 1). The specimens were sourced from the Anjialing North Gang sandstone block of China Coal Pingshuo, selected for its uniform texture and absence of significant joints, ensuring consistent rock integrity. These specimens were prepared into cylindrical shapes measuring 50 mm × 100 mm by 'cutting, grinding' processes. Fissures were prefabricated using waterjet cutting and oriented at angles of 0˚. 10˚, 30˚, 40˚, 10˚, 30˚, 40˚, 10˚, 30˚, and 40˚ with respect to the horizontal direction, as illustrated in Fig 2.

The experimental loading process employed a displacement control mode with a loading rate of $1\times10^{-3}$ mm/s and a data sampling interval of 1 s. The stress-strain curves for the rock specimens at various inclination angles were derived from the experiments, as depicted in Fig 3.

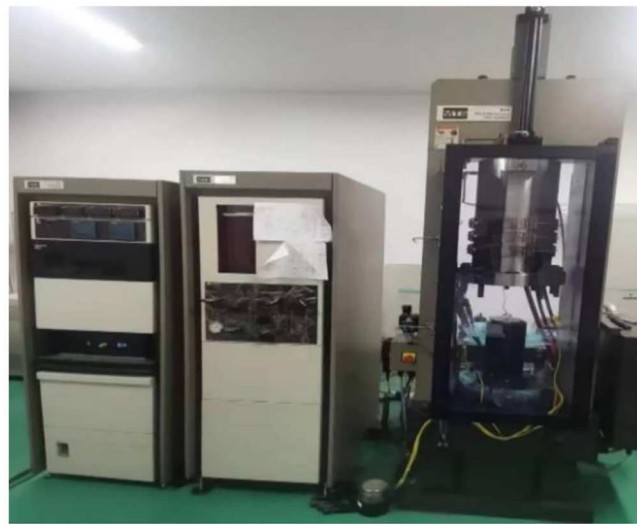

**Fig 1. MTS815 fully digital hydraulic servo test system.**

The stress-strain curves were comprehensively analyzed, revealing distinct phases, including compression-dense, elastic, yielding, and post-peak residual stages. Notably, no abrupt stress drop was observed in the late damage stage. Conversely, the stress can be stabilized basically at a certain value. Initially, during loading, owing to the closure of the original pore cracks in the rock specimens [16], the axial stress-strain curve exhibited an upward concave trend [17]. Subsequently, as the specimens transitioned into a continuous medium, they entered a linear elasticity phase until reaching the peak stress, followed by a notable stress drop [18] and entry into the residual strength stage. With increasing inclination of the structural surface, the proportion of the compaction phase in the rock specimens increased, whereas the proportion of the linear-elastic phase decreased, and post-peak instability development was weakened, and a high residual strength was preserved. This indicated the significant impact of the structural surface inclination on the mechanical properties of rock specimens.

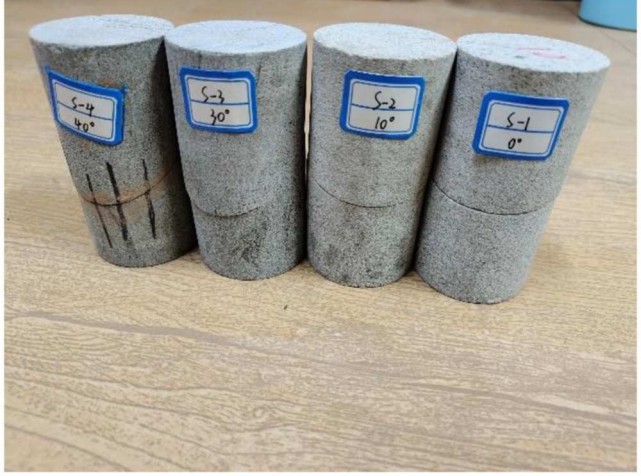

**Fig 2. Specimens of sandstone with different inclination angles.**

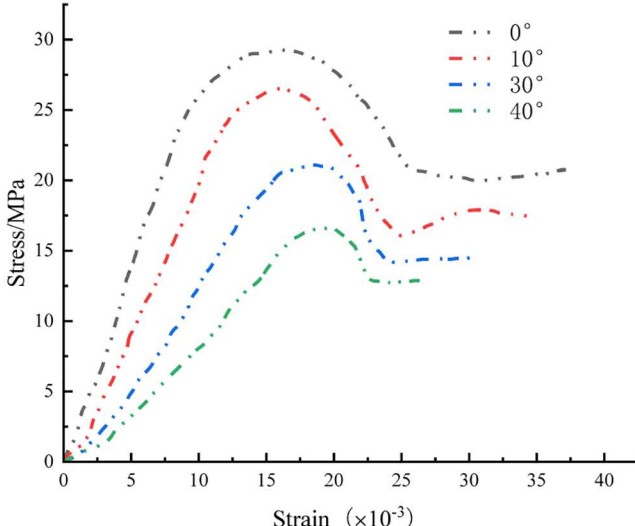

**Fig 3. Stress-strain curves of rock body with different inclination structural surfaces.**

## 2.2 Mechanical characteristics of rock specimens from structural surfaces with different inclinations

To accurately reflect the rupture effect of rock specimens with various inclined surfaces, three mechanical parameters were fitted, as shown in Figs 4–6.

An analysis of the experimental data revealed a positive correlation between the peak strain of the rock specimen and the inclination angle magnitude within the 0–40° range. Conversely, both the peak strain and modulus of elasticity exhibited negative correlations with the inclination-angle magnitude. Because of the high degree of curve fitting, the model employed three equations and six parameters to accurately represent the functional expressions of the peak

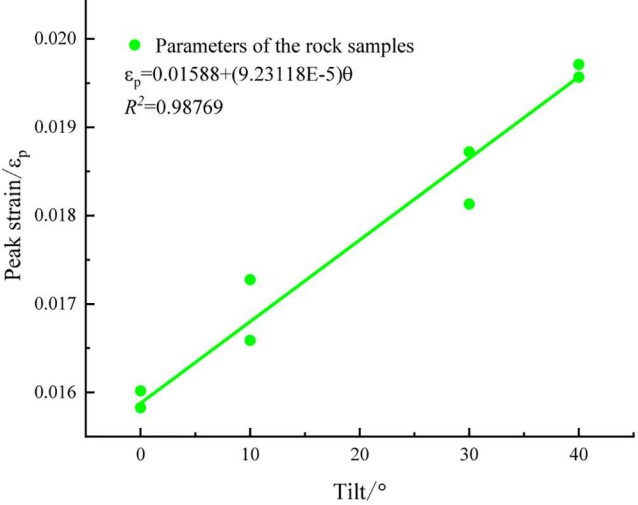

**Fig 4. Peak strain fitting curve.**

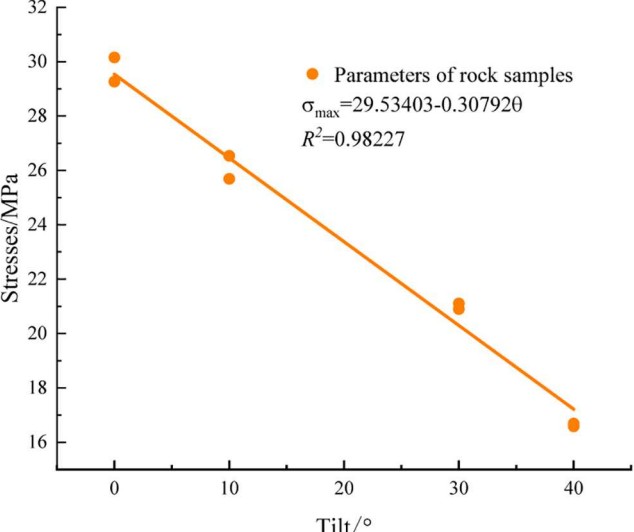

**Fig 5. Fitted curve of compressive strength.**

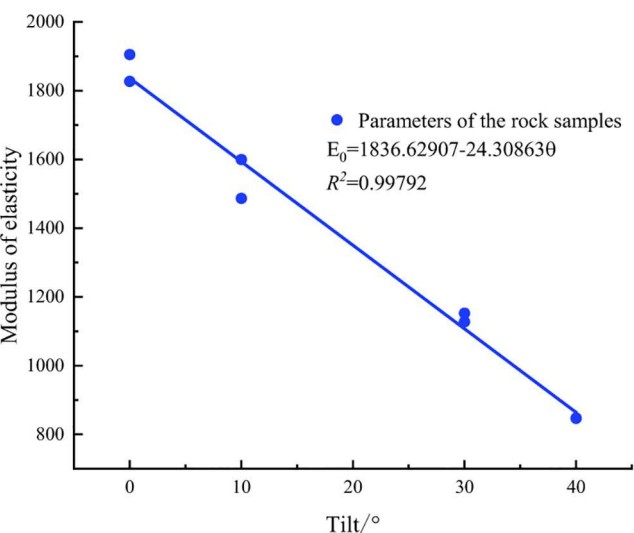

**Fig 6. Fitted curve of modulus of elasticity.**

strain, compressive strength, elastic modulus, and inclination angle [19].

$$\begin{cases} \varepsilon_p = a + b\theta \\ \sigma_{max} = c - d\theta \\ E_0 = f - g\theta \end{cases} \tag{1}$$

where a, b, c, d, f, and g represent the fitting parameters; $\varepsilon_p$ represents the peak strain of the specimen; $\sigma_{max}$ represents the compressive strength of the specimen; $E_0$ denotes the modulus of elasticity of the specimen; and $\theta$ denotes the angle of the structural surface of the specimen.

## 3 Energy evolution law of the deformation and destruction process of rock specimens

### 3.1 Principle of energy calculation

The deformation damage to rock specimens under external loading can be viewed as a dynamic evolutionary process involving energy input, accumulation, and dissipation. This process involves energy dissipation during the crack initiation, extension, and macro-expansion of fissures [20–22]. Hence, uncovering the impact mechanism of the structural surface inclination on rock mass damage and destruction is of significant importance.

According to the laws of thermodynamics, the energy U [23] produced by the work of external force is as follows.

$$U_0 = U_d + U_e \tag{2}$$

As shown in Fig 7, $U_d$ represents the pre-peak dissipated energy accumulated inside the rock specimen when the peak strength is reached, that is, the area in the figure is $S_{OABC}$; and $U_e$ denotes the elastic strain energy that can be released, that is, the area in the diagram is $S_{CBD}$.

Without considering the heat exchange effects, the energy produced by the work of the external force will be entirely absorbed by the rock specimen under uniaxial compression conditions [24, 25]:

$$\sigma_2 = \sigma_3 = 0 \tag{3}$$

$$U_0 = \int_0^{\varepsilon_1} \sigma_1 d\varepsilon_1 + \int_0^{\varepsilon_2} \sigma_2 d\varepsilon_2 + \int_0^{\varepsilon_3} \sigma_2 d\varepsilon_3 = \int_0^{\varepsilon_1} \sigma_1 d\varepsilon_1 \tag{4}$$

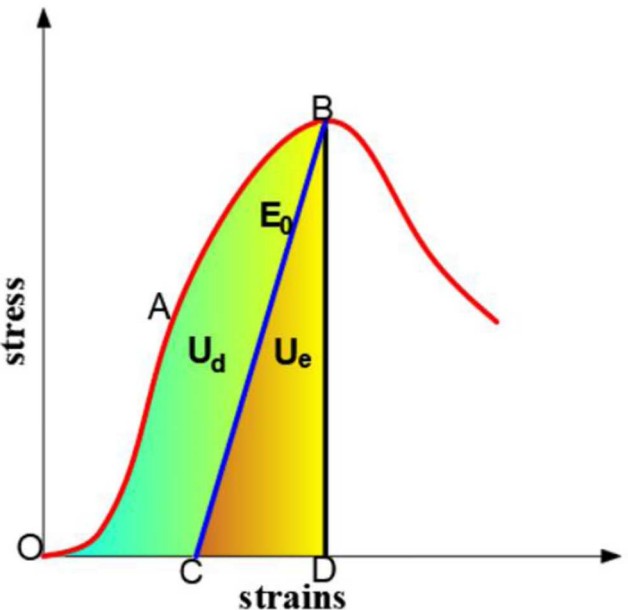

**Fig 7. Evolution of the whole energy process.**

$$U_e = \frac{1}{2}\sigma_1\varepsilon_1 = \frac{\sigma_1^2}{2E_u} \approx \frac{\sigma_1^2}{2E_0} \tag{5}$$

$$U_d = U_0 - U_e = \int_0^{\varepsilon_1} \sigma_1 d\varepsilon_1 - \frac{\sigma_1^2}{2E_0} \tag{6}$$

where $\sigma_1$, $\sigma_2$, and $\sigma_3$ represent the maximum, intermediate, and minimum principal stresses of the sample; $\varepsilon_1$, $\varepsilon_2$, and $\varepsilon_3$ represent the strains along the principal stresses; and $E_0$ denotes the actual modulus of elasticity of the sample.

## 3.2 Energy evolution law of rock specimens with different inclined structural surfaces

Based on the aforementioned principles, the energy evolution and stress-strain relationship of the rock body with various structural surface inclinations during the loading deformation damage process were computed. As depicted in Fig 8, the curve is delineated into four stages based on evolutionary laws.

1. Pressure-tight stage (OA): With increasing inclination of the structural surface, the compaction stage proportion of the rock specimens increased, resulting in a concave stress-strain curve. Concurrently, the total energy density, elastic strain energy density, and dissipation energy density exhibited slow increments. This phenomenon resulted from the presence of primary pores within the interior, where the external load predominantly closed the primary pores of the coal rock.

2. Resilience stage (AB): After the compaction of the rock specimens, the rock specimens exhibited notable linear elastic characteristics. The energy evolution diagram revealed a rapid increase in both the total energy and elastic strain energy density across the four groups of rock specimens. Primarily, the external energy absorbed by the rock specimens was converted into elastic strain energy [26, 27].

3. Yield stage (BC): The rock specimens exhibited irreversible plastic deformation under axial loading, as shown in Table 1. At 0° inclination, damage occurred at an elastic strain of 0.25192 MJ·m⁻³, with dissipation energy recorded at 0.05795 MJ·m⁻³, and the total strain energy was 0.30987 MJ·m⁻³. At a 10° inclination, damage manifested at an elastic strain of 0.2072 MJ·m⁻³, with a dissipation energy of 0.04398 MJ·m⁻³, yielding a total strain energy of 0.25118 MJ·m⁻³. Similarly, at 30° inclination, damage initiated at an elastic strain of 0.171 MJ·m⁻³, with a dissipation energy of 0.03378 MJ·m⁻³, totaling 0.20478 MJ·m⁻³ in strain energy. At a 40° inclination, damage began at an elastic strain of 0.1376 MJ·m⁻³, with dissipation energy of 0.03057 MJ·m⁻³, and the total strain energy was 0.16817 MJ·m⁻³. The percentages of energy dissipated relative to the total strain at the point of destruction were 18.7%, 17.51%, 16.5%, and 16.39%, respectively. Notably, this proportion decreased with increasing inclination of the structural plane, indicating a progressive increase in damage intensity within the rock specimens [28].

4. Post-peak residual stage (CD): After the macroscopic destruction of the rock specimen, the stress rapidly declined and stabilized at a specific value. The elastic strain energy accumulated during the elastic phase was rapidly discharged [29], and the dissipated energy density rapidly increased.

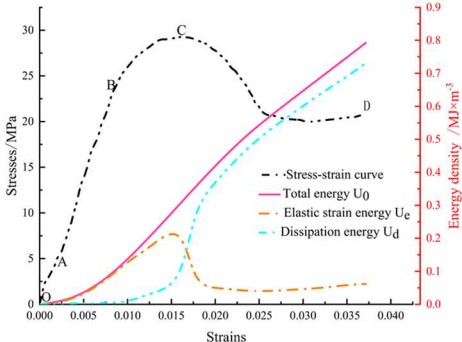

(a) Plot of the energy evolution pattern of a 0° dipping rock mass

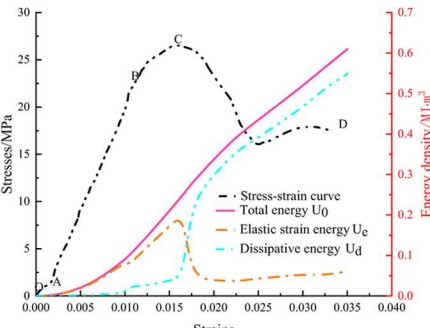

(b) Plot of the energy evolution pattern of a 10° dipping rock mass

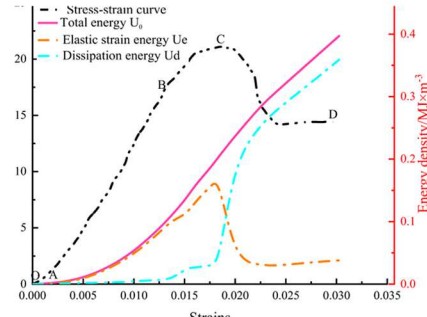

(c) Plot of the energy evolution pattern of a 30° dipping rock mass

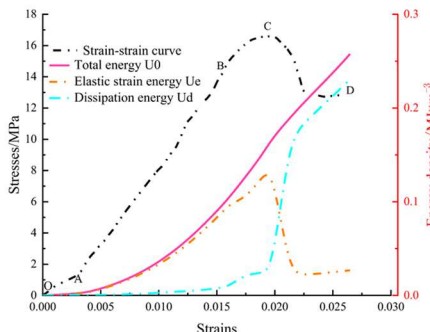

(d) Plot of energy evolution pattern of 40° dipping rock mass

**Fig 8. Plot of energy evolution pattern of rock samples with different inclination angles.**

**Table 1. Energy trends at the peak of structural surfaces in sandstones with different dip angles.**

| rock specimen | Structural surface inclination θ (°) | Total strain energy U0 at destruction (MJ·m⁻³) | Elastic strain energy Ue at peak (MJ·m⁻³) | Dissipated energy Ud at peak (MJ·m⁻³) |
|---|---|---|---|---|
| Sandstone S-1 | 0 | 0.30987 | 0.25192 | 0.05795 |
| Sandstone S-2 | 10 | 0.25118 | 0.2072 | 0.04398 |
| Sandstone S-3 | 30 | 0.20478 | 0.171 | 0.03378 |
| Sandstone S-4 | 40 | 0.16817 | 0.1376 | 0.02757 |

In summary, for inclination angles less than 40°, both the elastic strain energy and total strain energy at the rock specimen damage decreased with increasing inclination angle of the structural plane. This suggested that the increase in the inclination angle of the structural plane of the rock specimens reduced their ability during the online elasticity stage to store elastic strain energy and absorb external energy. Dissipated energy served as the energy source for rock specimen damage and was inevitably accompanied by energy dissipation. With increasing structural plane inclination angle, the dissipated energy at the peak of the sample decreased, and the proportion of dissipated energy decreased accordingly.

## 4 Damage evolution model for rock specimens based on minimum energy dissipation principle

### 4.1 Ontological model for damage evolution of structural surfaces with different inclination angles

The destruction of rock specimens involves energy conversion, which is governed by the principle of minimum energy consumption. This principle suggests that any energy-consuming process can be conducted in the least energy-consuming manner, within its constraints. The "corresponding constraints" denotes the control equations and solution conditions necessary for the physical quantities in the energy consumption rate expression of the energy-consuming process. The "in the least energy-consuming manner" indicates that at any point in the energy consumption process, its energy consumption rate was the minimum among all energy depletion rates [30].

Since this experiment was uniaxial loading, the perimeter pressure presented $\sigma_2 = \sigma_3 = 0$. Therefore, the ontological relationship of the rock specimens before damage is as follows.

$$\varepsilon = \frac{\sigma}{E} \tag{7}$$

According to the strain equivalence principle, Eq (7) can be represented as

$$\varepsilon = \frac{\sigma}{[1 - D(t)]E} \tag{8}$$

When the axial stress is applied, the rate of loss due to D(t) is given by

$$\varphi(t) = \sigma_i \dot{\varepsilon}_i^N(t) \tag{9}$$

where $\varepsilon_i^N(t)$ is the strain rate, and t is the damage time. From the above equation, we obtain:

$$\varepsilon = \frac{-\dot{D}(t)}{[1 - D(t)]E}\sigma \tag{10}$$

Substituting Eq (10) into Eq (9) yields

$$\varphi(t) = -\frac{\dot{D}(t)}{[1 - D(t)]^2 E}\sigma^2 \tag{11}$$

The constraints on the rock specimen in the energy-dissipation process can be expressed as follows:

$$F(\sigma) = \sigma - R_c = 0 \tag{12}$$

where $R_c$ is the compressive strength of the rock specimen.

According to the principle of minimum energy dissipation, Eq (11) conforms to the requirement of attaining the stationary value given the condition specified by Eq (12), which is achieved through the introduction of the Lagrange multiplier λ:

$$\frac{\partial[\varphi(t) + \lambda F]}{\partial \sigma} = 0 \tag{13}$$

This can be obtained using the following damage mechanics:

$$\sigma = E(1 - D)\varepsilon \tag{14}$$

This is obtained by combining Eqs (12)–(14), respectively.

$$D(t) = 1 - e^{\left(\frac{\lambda}{2\varepsilon} + c\right)} \tag{15}$$

As shown in Fig 3,

$$\begin{cases} \varepsilon = \varepsilon_P, \sigma = \sigma_{\max} \\ \varepsilon = \varepsilon_P, \dfrac{d\sigma}{d\varepsilon} = 0 \end{cases} \tag{16}$$

where $\varepsilon_p$ is the peak strain, and $\sigma_{\max}$ is the peak stress. By calculating, we obtain

$$\lambda = 2\varepsilon_p \tag{17}$$

$$c = \ln \sigma_{\max} - \ln Ee\varepsilon_p \tag{18}$$

By substituting λ and c values into Eq (15) leads to the damaged ontological model of the rock specimen as follows:

$$D(t) = 1 - e^{\left(\frac{\varepsilon_p}{\varepsilon} + \ln \sigma_{\max} - \ln Ee\varepsilon_p\right)} \tag{19}$$

By substituting Eq (1) into Eq (19), the damage ontological model for rock specimens with different inclination structural surfaces was introduced as follows:

$$D(t) = 1 - \exp\{\frac{a + b\theta}{\varepsilon} + \ln(c - d\theta) - \ln[(f - g\theta)e(a + b\theta)]\} \tag{20}$$

## 4.2 Determine the damage threshold and critical value

When no damage occurred in the rock specimen,

$$D(t) = 1 - \exp(\frac{\lambda}{2\varepsilon} + c) = 0 \tag{21}$$

That is, the rock specimen damage threshold is

$$\varepsilon_0 = -\frac{\lambda}{2c} = \frac{\varepsilon_p}{\ln(E_0 e \varepsilon_p) - \ln(\sigma_{\max})} \tag{22}$$

Obviously, $\varepsilon_0 < \varepsilon_p$. Hence, the rock specimens were damaged before the damage stage, which was corroborated by the critical behavior observed in the stress-strain relationship curve.

Damage thresholds for rock specimens with different inclinations when $\varepsilon_0 < \varepsilon_p$:

$$D(t) = 1 - \exp[\ln(\sigma_{\max}) - \ln(E_0 e \varepsilon_p)] \tag{23}$$

## 4.3 Damage evolution curves for rock specimens with different inclination angles

The fitted parameter $a = 0.01588$, $b = 9.23118 \times 10^{-5}$, $c = 29.53403$, $d = 0.30792$, $f = 1836.62907$, and $g = 24.30863$ are substituted into Eq (20), and the curve is plotted through MATLAB. As depicted in Fig 9.

The axial strain of the rock specimens with varying inclination angles reached the damage threshold at a specific loading level. Subsequently, the specimens progressed into the fissure development stage, with damage initiation occurring after the axial strain surpassed the threshold. This progression was represented by an upward concave curve. The damage thresholds for the samples at inclination angles of 0˚, 10˚, 30˚, and 40˚ were 0.01542, 0.01564, 0.01634, and 0.01682, respectively. These thresholds demonstrated increases of 1.43%, 5.97%, and 9.08%, respectively, compared to the 0˚ inclination angle samples. Correspondingly, the damages critical for the structural faces at these angles were 0.6086, 0.6538, 0.6757, and 0.692, respectively. These values demonstrated increases of 7.43%, 11.03%, and 13.7%, respectively, compared to the structural face at 0˚ inclination. In an ideal scenario, when the bearing capacity of the rock specimen was entirely depleted owing to the applied load, its effective bearing area should be 0, corresponding to a damage variable of 1. However, the analysis of the curves revealed that specimens with various inclination angles tended to stabilize upon reaching a

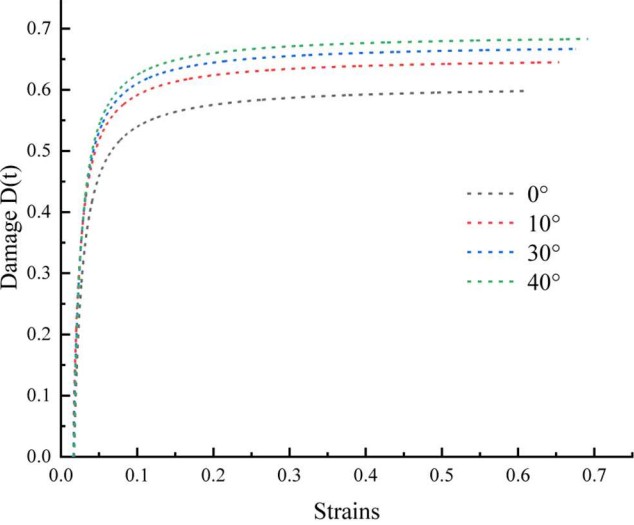

**Fig 9. Damage evolution curves of rock samples with different inclination angles.**

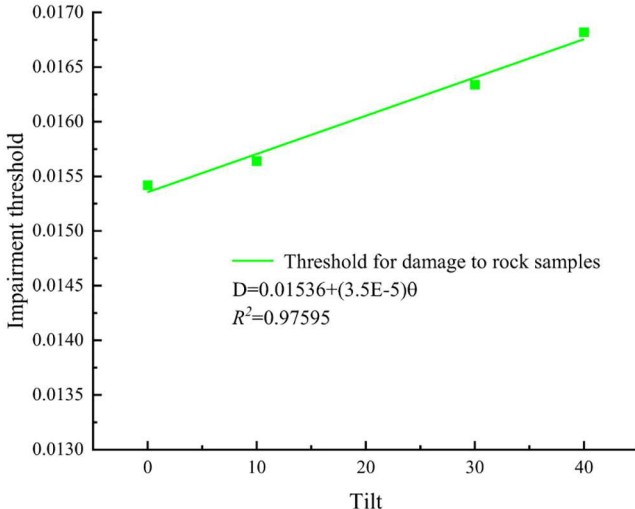

**Fig 10. Trend of damage threshold for rock samples with different inclination angles.**

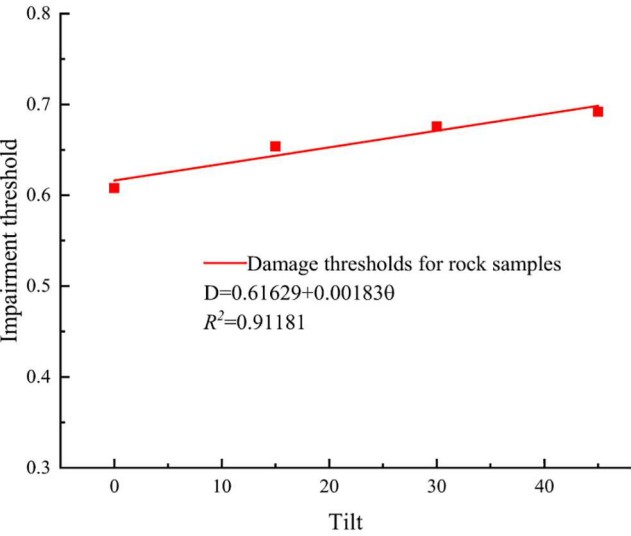

**Fig 11. Trend of damage thresholds of rock samples with different inclination angles.**

critical value range, typically below 1. This observation suggested that even when fully damaged, the rock specimens maintained some residual bearing capacity.

The damage thresholds and critical damage values of rock specimens with different inclination angles were fitted, as shown in Figs 10 and 11.

The analysis of Figs 10 and 11 revealed that as long as the inclination angle of the structural surface of the rock specimens remained below 40˚, both the damage threshold value and the critical damage value of the specimens exhibited a proportional increase with the inclination angle. This suggested that variations in the inclination angle of the structural surface affected the propagation of internal cracks in the rock specimens under external loading. Combined with the stress-strain curves, at equivalent axial loads, greater inclination angles corresponded

to higher damage threshold values in the rock specimens. This indicated a delayed onset of plastic development in these specimens. The primary factor behind this observation was the alteration of the stress state during the compression-densification stage induced by changes in the inclination angle of the structural plane, increasing both the damage threshold value and the damage threshold value. The primary cause of this phenomenon was the alteration of the stress state in the compression-densification stage of the rock specimens induced by the inclination of the structural plane, thereby promoting the initiation and propagation of internal cracks.

From Eq (23), the critical damage value was related to the three mechanical parameters of peak strain, compressive strength, and elastic modulus, and was a comprehensive index of these three parameters.

## 5 Conclusion

1. Under axial loading, after the attainment of the peak stress, the rock specimen entered an unstable development stage. This stage was gradually weakened as the inclination angle of the structural surface remained below 40˚. Moreover, the peak strain decreased, while both the peak strain and modulus of elasticity increased as the inclination angle increased.

2. As the inclination angle of the structural plane increased, the capacity of the rock specimens to store elastic strain energy and absorb external energy during the initial elastic phase decreased. Energy dissipation that was inherently accompanied by its development was the source of coal rock damage. As the inclination angle of the structural surface increased, both the dissipative energy at the peak of the rock specimens and its proportion steadily decreased.

3. The critical damage value of the rock specimen correlated with its compressive strength, peak strain, and elastic modulus, serving as a comprehensive indicator of these factors. Furthermore, when the inclination angle of the structural surface of the rock specimen remained below 40˚, it exhibited a positive correlation with both the damage threshold value and damage critical value.

4. This can offer valuable insight for the mitigation of geological hazards, such as collapse and sheeting, in rock formations featuring various inclined structural surfaces at depth.

## Supporting information

**S1 Data.**
(XLSX)

## Author Contributions

**Data curation:** Jiaming Liu.

**Software:** Yuntao Yang, Xu Dong.

**Writing – original draft:** Yongjiang Yu, Zhiyuan Song.

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
