## [Decision Letter · Decision Letter 0]

4 Feb 2024

PONE-D-24-01554Energy evolution mechanism of structural surfaces in sandstones with different dips based on the energy principlePLOS ONE

Dear Dr. Zhiyuan,

Thank you for submitting your manuscript to PLOS ONE. After careful consideration, we feel that it has merit but does not fully meet PLOS ONE’s publication criteria as it currently stands. Therefore, we invite you to submit a revised version of the manuscript that addresses the points raised during the review process.

We look forward to receiving your revised manuscript.

Kind regards,

Kang Wang, Ph.D.

Academic Editor

PLOS ONE

Journal Requirements:

3. We note that your Data Availability Statement is currently as follows: "All relevant data is contained in the manuscript and its supporting information file"

Reviewers' comments:

Reviewer's Responses to Questions

**Comments to the Author**

1. Is the manuscript technically sound, and do the data support the conclusions?

Reviewer #1: Yes

Reviewer #2: Yes

2. Has the statistical analysis been performed appropriately and rigorously? 

Reviewer #1: Yes

Reviewer #2: Yes

3. Have the authors made all data underlying the findings in their manuscript fully available?

Reviewer #1: Yes

Reviewer #2: Yes

4. Is the manuscript presented in an intelligible fashion and written in standard English?

Reviewer #1: Yes

Reviewer #2: Yes

5. Review Comments to the Author

Reviewer #1: In this paper, uniaxial compression tests are conducted on sandstones with different inclined structural surfaces to analyze their deformation characteristics and energy evolution laws in the deformation and damage process; finally, the damage constitutive model of nodular sandstones is established based on the principle of minimum energy consumption. The results of the study have certain theoretical value for enriching the mechanical model of jointed rock. However, some questions in the article need to be modified or explained by the authors, as follows:

1. Most of the research results of scholars listed in the introductory part of the paper are the results of mechanical tests on rock bodies containing structural surfaces, and there are relatively few relevant results involving energy evolution; in addition, the relevant results of the principle of minimum energy consumption also need to be listed.

2 In Section 2.2, the author should describe in detail the source of rock specimens.

3. The colors of the stress-strain curves in Fig. 8 for plots a, b, c, and d are not uniform. It is recommended that the authors re-plot the curves in the text.

4. Some references are not cited correctly, and the writing format and English expression should follow the standards of the International Journal.

5. The language of this manuscript is not high enough and there are many unclear expressions and grammatical errors. Authors should improve the quality of the language.

Reviewer #2: The topic sounds practically useful but some revisions should be made as follows:

1.Please add the line number.

2.The objectives and the rationale of the study should be clearly stated.

3.The background of the problem can be presented in the introduction.

4.Would you mind providing an explanation for Ue and Ud in Figure 7? Additionally, could you specify the citation for Figure 7 in the manuscript?

5.How was the equation 1 derived, and could you please provide a more detailed explanation for its derivation?

6.How many rock samples have you analyzed?

7.“The damage threshold of the rock sample is only related to its compressive strength, peak strain, and elastic modulus, and is a comprehensive index of all three”. You mentioned the term 'only' in the conclusion. Could you elaborate on how you arrived at these conclusions.

8.“In the process of deep mining, appropriate protective measures shall be taken for rock bodies containing structural surfaces with different inclinations, in order to prevent geological hazards such as collapses and sheet gangs from occurring in the rock bodies.” The stress state in deep mining is complex, and relying solely on uniaxial compressive tests may be insufficient. How did you come to this conclusion?

9.Most of the references are relatively outdated, and some of the latest research related to rock mechanisms can be added, such as: 10.1016/j.fuel.2023.129584, 10.1016/j.compgeo.2024.106095.

10.The abstract is lengthy; kindly condense it.

6. PLOS authors have the option to publish the peer review history of their article (what does this mean?). If published, this will include your full peer review and any attached files.

Reviewer #1: No

Reviewer #2: No

---

## [Author Response · Author response to Decision Letter 0]

3 Mar 2024

Reviewer #1:

1. Comment: 1. Most of the research results of scholars listed in the introductory part of the paper are the results of mechanical tests on rock bodies containing structural surfaces, and there are relatively few relevant results involving energy evolution; in addition, the relevant results of the principle of minimum energy consumption also need to be listed.

Response: Thanks for reviewer’s comment. The authors have already listed energy evolution and minimum energy consumption-related results in the introduction. 

2. Comment: In Section 2.2, the author should describe in detail the source of rock specimens.

Response: Thanks for reviewer’s comment. The authors have described in detail the origin of the rock specimens. 

3. Comment: The colors of the stress-strain curves in Fig. 8 for plots a, b, c, and d are not uniform. It is recommended that the authors re-plot the curves in the text.

Response: Thanks for reviewer’s comment. The different colored stress-strain curves in Fig. 8 correspond to Fig. 3, but this does detract from the aesthetics and I have standardized them. 

4. Comment: Some references are not cited correctly, and the writing format and English expression should follow the standards of the International Journal.

Response: Thanks for reviewer’s comment. The author has revised it carefully. 

5. Comment: The language of this manuscript is not high enough and there are many unclear expressions and grammatical errors. Authors should improve the quality of the language.

Response: Thanks for reviewer’s comment. The manuscript has been referred to professionals for touch-ups and various inappropriate expressions and grammatical issues have been corrected. 

Reviewer #2:

1. Comment: Please add the line number.

Response: Thanks for reviewer’s comment. The author has added line numbers to the text. 

2. Comment: The objectives and the rationale of the study should be clearly stated.

Response: Thanks for reviewer’s comment. The authors have stated at the end of the abstract.

3. Comment: The background of the problem can be presented in the introduction.

Response: Thanks for reviewer’s comment. The authors have provided an introduction.

4. Comment: Would you mind providing an explanation for Ue and Ud in Figure 7? Additionally, could you specify the citation for Figure 7 in the manuscript?

Response: Thanks for reviewer’s comment. Consider a rock unit deformed by an external force, assuming that there is no heat exchange between the physical process and the outside world, i.e., a closed system and that the total input energy from the external work is U. From the first law of thermodynamics, we have U = Ud + Ue, where Ud is the dissipated energy and Ue is the releasable elastic strain energy. The authors have already explained Ue and Ud in the text and pointed out the references to Fig. 7. 

5. Comment: How was equation 1 derived, and could you please provide a more detailed explanation for its derivation?

Response: Thanks for reviewer’s comment. As shown in Figures 4, 5, and 6, due to the good fit of the experimental data, the fitted three curve equations derived from Origin were then linked as a function of peak strain, compressive strength, and modulus of elasticity concerning the inclination angle, and a, b, c, d, f, and g were used as the fitted parameters in place of the specific values in the equations, respectively. 

6. Comment: How many rock samples have you analyzed?

Response: Thanks for reviewer’s comment. The rock samples were all cut from the sandstone blocks of the Anjialing North Gang slope of China Coal Pingshuo, and 3 rock samples were processed for each inclination, totaling 12 rock samples, and due to the influence of space, only the most representative group was selected and put into the text.

7. Comment: The damage threshold of the rock sample is only related to its compressive strength, peak strain, and elastic modulus, and is a comprehensive index of all three”. You mentioned the term 'only' in the conclusion. Could you elaborate on how you arrived at these conclusions?

Response: Thanks for reviewer’s comment. As can be seen from the derived Eq. (23), the only parameters that affect the damage threshold in this paper are compressive strength, peak strain, and modulus of elasticity, but the word "only" has been deleted from the manuscript because it is indeed too absolute.

8. Comment: In the process of deep mining, appropriate protective measures shall be taken for rock bodies containing structural surfaces with different inclinations, in order to prevent geological hazards such as collapses and sheet gangs from occurring in the rock bodies.” The stress state in deep mining is complex, and relying solely on uniaxial compressive tests may be insufficient. How did you come to this conclusion?

Response: Thanks for reviewer’s comment. In the early review of the literature, the authors found that many scholars for the deep rock body force analysis also from the uniaxial compression experiment, but as you said, the deep mining stress state is complex, only uniaxial compression test is not enough, so the authors of the original conclusion were modified, and will be used in the subsequent study of triaxial compression test to continue to analyze the mechanical properties of the structural surface of the different inclination angle. 

9. Comment: Most of the references are relatively outdated, and some of the latest research related to rock mechanisms can be added, such as 10.1016/j.fuel.2023.129584, 10.1016/j.compgeo.2024.106095.

Response: Thanks for reviewer’s comment. The authors have added relevant and up-to-date research findings.

10. Comment: The abstract is lengthy; kindly condense it.

Response: Thanks for reviewer’s comment. The authors have streamlined some of the abstracts.

---

## [Decision Letter · Decision Letter 1]

6 Mar 2024

Energy evolution mechanism of structural surfaces in sandstones with different dips based on the energy principle

PONE-D-24-01554R1

Dear Dr. Zhiyuan,

We’re pleased to inform you that your manuscript has been judged scientifically suitable for publication and will be formally accepted for publication once it meets all outstanding technical requirements.

Kind regards,

Kang Wang, Ph.D.

Academic Editor

PLOS ONE

Additional Editor Comments (optional):

Reviewers' comments:

Reviewer's Responses to Questions

**Comments to the Author**

1. If the authors have adequately addressed your comments raised in a previous round of review and you feel that this manuscript is now acceptable for publication, you may indicate that here to bypass the “Comments to the Author” section, enter your conflict of interest statement in the “Confidential to Editor” section, and submit your "Accept" recommendation.

Reviewer #2: All comments have been addressed

2. Is the manuscript technically sound, and do the data support the conclusions?

Reviewer #2: Yes

3. Has the statistical analysis been performed appropriately and rigorously? 

Reviewer #2: Yes

4. Have the authors made all data underlying the findings in their manuscript fully available?

Reviewer #2: Yes

5. Is the manuscript presented in an intelligible fashion and written in standard English?

Reviewer #2: Yes

6. Review Comments to the Author

Reviewer #2: (No Response)

7. PLOS authors have the option to publish the peer review history of their article (what does this mean?). If published, this will include your full peer review and any attached files.

Reviewer #2: **Yes: **lizhaoxin

---

## [Editor Report · Acceptance letter]

11 Mar 2024

PONE-D-24-01554R1 

PLOS ONE

Dear Dr. Song, 

I'm pleased to inform you that your manuscript has been deemed suitable for publication in PLOS ONE. Congratulations! Your manuscript is now being handed over to our production team.

Kind regards, 

on behalf of

Dr. Kang Wang 

Academic Editor

PLOS ONE